# Systematic Review of the Short-Term versus Long-Term Duration of Antibiotic Management for Neutropenic Fever in Patients with Cancer

**DOI:** 10.3390/cancers15051611

**Published:** 2023-03-05

**Authors:** Kazuhiro Ishikawa, Tetsuhiro Masaki, Fujimi Kawai, Erika Ota, Nobuyoshi Mori

**Affiliations:** 1Department of Infectious Diseases, St. Luke’s International Hospital, Tokyo 104-8560, Japan; 2Library, Center for Academic Resources, St. Luke’s International University, Tokyo 104-0044, Japan; 3Global Health Nursing, Graduate School of Nursing Sciences, St. Luke’s International University, Tokyo 104-0044, Japan; 4Tokyo Foundation for Policy Research, Tokyo 106-6234, Japan

**Keywords:** febrile neutropenia, short-term duration of antibiotics, cancer patient

## Abstract

**Simple Summary:**

Empirical administration of broad-spectrum antibiotics during neutropenia has been shown to reduce mortality from bacterial infections. However, prolonged antibiotic exposure, in particular, promotes the development of antimicrobial resistance and the selection of resistant microorganisms, which are often more difficult to treat, and carry a higher risk of complications. Early antibiotic discontinuation has been proposed in patients with hematologic malignancy who have febrile neutropenia. Several studies have found that shorter duration of antimicrobial therapy have better clinical outcomes and lower the exposure to the broad-spectrum antibiotics, but this raises concerns about their implementation in clinical practice. Furthermore, their safety and efficacy have been questioned. In our study, a systematic review was conducted to compare the short-term and long-term durations of antibiotics for febrile neutropenia for the outcomes of clinical failure, mortality, and bacteremia.

**Abstract:**

Early antibiotic discontinuation has been proposed in patients with hematologic malignancy with fever of unknown origin during febrile neutropenia (FN). We intended to investigate the safety of early antibiotic discontinuation in FN. Two reviewers independently searched for articles from Embase, CENTRAL, and MEDLINE on 30 September 2022. The selection criteria were randomized control trials (RCTs) comparing short- and long-term durations for FN in cancer patients, and evaluating mortality, clinical failure, and bacteremia. Risk ratios (RRs) with 95% confidence intervals (CIs) were calculated. We identified eleven RCTs (comprising 1128 distinct patients with FN) from 1977 to 2022. A low certainty of evidence was observed, and no significant differences in mortality (RR 1.43, 95% CI, 0.81, 2.53, I^2^ = 0), clinical failure (RR 1.14, 95% CI, 0.86, 1.49, I^2^ = 25), or bacteremia (RR 1.32, 95% CI, 0.87, 2.01, I^2^ = 34) were identified, indicating that the efficacy of short-term treatment may not differ statistically from that of long-term treatment. Regarding patients with FN, our findings provide weak conclusions regarding the safety and efficacy of antimicrobial discontinuation prior to neutropenia resolution.

## 1. Introduction

Fever due to chemotherapy-induced neutropenia is experienced in 10–50% of patients with solid tumors, and more than 80% of those with hematologic malignancies [1]. Patients with hematological malignancy are at high risk of febrile neutropenia (FN), and experience Gram-negative bacilli bloodstream infections. Broad-spectrum beta-lactam antibiotics should be administered, such as carbapenem, piperacillin/tazobactam, ceftazidime, or cefepime, according to several guidelines [1,2,3]. Determining the optimal duration of empirical antimicrobial therapy for patients with hematological malignancies, including those requiring stem cell transplantation, solid tumors, and FN of unknown origin, remains challenging.

The Infectious Diseases Society of America [1] and the European Conference on Infections in Leukemia [2] recommend different strategies for the management of FN of unknown origin, such as early de-escalation of antibiotics from empiric to prophylactic therapy until the recovery of neutrophils, or the complete cessation of antibiotics. In two large clinical trials of empirical antimicrobial therapy with hematological malignancies and FN [4,5], the mean duration of treatment was longer than the recommended duration of 8 days or less [6] for treatment of the most serious infections. Approximately 70% of neutropenic fevers are classified as fever of unknown origin, and antimicrobial therapy may be unnecessary [7]. Anaerobic antibiotics, such as piperacillin/tazobactam or carbapenem, are associated with an increased risk of acute gut/liver graft-versus-host disease in stem cell transplantation [8,9].

In general, a shorter duration of antimicrobial therapy reduces the development of multidrug-resistant organisms; incidence of antimicrobial-induced adverse drug events, such as antibiotic-associated diarrhea, *Clostridium difficile* colitis, allergy reaction, and nephrotoxicity; and hospital length of stay. In hematological cancer patients, the reduction of antimicrobial therapy durations lead to less unnecessary antimicrobial exposure, which is an additional benefit for patients with hematological cancer that has not previously been described in this vulnerable population. For example, treatment of FN with imipenem-cilastatin and piperacillin-tazobactam antibiotics was associated with increased GVHD-related mortality at 5 years, whereas aztreonam and cefepime were not associated with GVHD-related mortality [10].

This discovery may also help in the development of antimicrobial stewardship programs, which are crucial for patients with hematological cancers who are repeatedly exposed to broad-spectrum antimicrobials for infection prevention and treatment [11].

However, the clinical practice for antimicrobial management in the setting of FN with unknown origin is under discussion. de Jonge et al. [12] reported no significant differences in clinical failure, safety, or mortality with early discontinuation of antibiotics, compared with that of continuation of antibiotics until neutrophil recovery. This study had a very strong impact on the de-escalation strategy in FN. On the other hand, Stern et al. [13] reported no significant difference in clinical failure between the short- and long-term antibiotics treatment groups in a systematic review until 2018 (risk ratio [RR] 1.23, 95% confidence interval [CI] 0.85–1.77), but the evidence was of low quality.

We aimed to conduct a systematic review of clinical outcomes between short- and long-term use of beta-lactam antibiotics in patients with high-risk FN, adding several new studies until 30 September 2022. We updated the systematic review with new evidence that had accumulated since the first version of the Stern et al. [13] publication, which revealed no significant differences in terms of outcomes or subsequent recommendations. We assessed the safety of short-term discontinuation of antibiotics, regardless of neutrophil count, compared to the long-term discontinuation of antibiotics until neutropenia resolution in people with fever and neutropenia, in terms of mortality, clinical failure, and bacteremia.

## 2. Materials and Methods

### 2.1. Objectives

To compare the safety (mortality, clinical failure, and bacteremia) of short- versus long-term beta-lactam antibiotic therapy in patients with febrile neutropenic cancer.

### 2.2. Sources and Searches

An investigator (F.K.) developed a search strategy. Three databases, namely Embase, the Cochrane Central Register of Controlled Trials (CENTRAL) in the Cochrane Library, and MEDLINE via Ovid, were searched until 30 September 2022 using similar search terms to those used in the previous systematic review by Stern et al. [13]. Each search strategy is shown in the Appendix A. This systematic review was conducted according to the guidelines of the Preferred Reporting Items for Systematic reviews and Meta-Analyses (PRISMA) [14]. The review protocol was recorded on 2 November 2022 with PROSPERO with the CRD number 42022369590.

### 2.3. Selection of Studies

We included randomized control trials (RCTs) in any language that reported all-cause mortality, clinical failure, or bacteremia, comparing the short-term duration of antibiotics with the long-term duration in hematological FN. We excluded patients with clinically and microbiologically documented infections, as well as neonatal patients. Two investigators (K.I. and T.M.) independently assessed the full texts of the articles. Discrepancies were discussed with a third and fourth investigator (E.O. and N.M.). Regarding the selection of studies until the end of 2017, we referred to the results of a Cochrane review by Stern et al. [13].

Adults (older than 18 years) and children (younger than 18 years) with FN caused by cancer chemotherapy and treated with any antibiotic regimen were included in this study. We defined fever as a single oral temperature higher than 38.3 °C or a temperature higher than 38.0 °C sustained for more than 1 h, according to the guidelines [1,2]. Neutropenia was defined as an absolute neutrophil count of less than 500 cells/μL. Studies that used a different although similar definition to that in the guidelines were included in the review. The types of interventions in the RCT define protocol-guided antibiotic discontinuation prior to neutropenia resolution versus antibiotic continuation until neutropenia resolution. We recorded the criteria defined for antibiotic discontinuation, including the timing of discontinuation, definitions of defervescence, and neutrophil count defined for neutropenia resolution.

### 2.4. Outcomes

The primary outcomes in this systematic review were any cause mortality, clinical failure, and bacteremia. Clinical failure was assessed as defined in each study.

### 2.5. Data Extraction

Two investigators (K.I. and T.M.) independently extracted the following data: publication country, published year, sample size, type of cancer (solid tumor or hematological malignancy, including stem cell transplantation), type of beta-lactam antibiotics, follow-up period, mortality, clinical failure, and bacteremia in each study. Data were extracted to the data extraction sheet using Microsoft Excel and Google spreadsheets, and were easily checked by a reviewer, including study information (e.g., publication country, study years, single-center or multi-center study), participant baseline characteristics (type of population, inclusion and exclusion criteria, comorbidity, and type of cancer), information regarding the intervention (type of antimicrobials and planned antibiotic duration in each arm), information regarding risk of bias (e.g., randomization method, allocation concealment, blinding, discontinuation of study, and incomplete outcome reporting), and information regarding outcomes (mortality, clinical failure, and bacteremia). Two review authors (K.I. and T.M.) extracted data from the included trials independently and entered them into the data extraction sheet. We extracted data preferentially using the intention-to-treat method, which included all individuals who were randomly assigned to the study outcome. For dichotomous outcomes, we recorded the number of participants manifesting the outcome in each group, as well as the number of evaluated participants. For continuous outcomes, we documented values, as well as the measure used, to represent the data (including mean with standard deviation and median with interquartile range). Discrepancies were resolved through discussion or by other investigators (E.O. and N.M.). We asked study authors for any missing data so that we could include findings from any studies published after 2018 in our study. We also referred to the data from the Stern et al. study [13], which was published before 2018.

### 2.6. Risk of Bias Assessment

Two investigators (K.I. and T.M.) independently assessed the risk of bias. Disagreements were resolved via discussion with a third investigator (E.O. and N.M.). The risk of bias was assessed according to the scales of the Cochrane risk-of-bias tool (RoB) [15]. With the RoB, we evaluated seven domains of bias: random sequence generation, allocation concealment, blinding of participants and personnel, blinding of outcome assessment, incomplete outcome data, reporting selection, and others. We assessed the effect of allocation concealment on results based on the evidence of a strong association between poor allocation concealment and overestimation of effect [16], as defined below:Low risk of bias (adequate allocation concealment);Unclear bias (uncertainty regarding allocation concealment);High risk of bias (inadequate allocation concealment).

The two review authors independently recorded methods of allocation generation, blinding, incomplete outcome data, selective reporting, the unit of randomization (patient or febrile episode), and publication status, in addition to the adequacy of allocation concealment.

### 2.7. Statistical Analyses

We analyzed dichotomous data by calculating the RR for each study, with the uncertainty in each result presented as 95% CIs. We assessed the percentage of variation across studies that could not be ascribed to sampling variation using the I^2^ statistic. A fixed-effects model was used unless significant heterogeneity was observed (*p* < 0.1 or I^2^ > 50%), in which case the random-effects model was used. We also visually inspected the forest plots to judge heterogeneity. We analyzed the data using Review Manager 5.4 (freely available software, released by Cochrane, London, UK).

### 2.8. Certainty of Evidence

We used the Grades of Recommendations, Assessment, Development, and Evaluations (GRADE) approach to interpret the findings and rate the certainty of evidence [17], grading the major outcomes (mortality, clinical failure, and bacteremia development). A certainty of evidence of review was evaluated using GRADEpro guideline development tool software (GRADEpro GDT, Evidence Prime Inc., Hamilton, ON, Canada) [18], using parameters such as study design, risk of bias, directness of outcomes, heterogeneity, precision within results, bias due to publication, estimate effect, and dose relationship with response and confounders. Thus, the overall GRADE obtained can be high, moderate, low, or very low certainty of evidence. We considered this analysis in our conclusions.

### 2.9. Sensitivity Analysis

We conducted a sensitivity analysis to assess the effect of allocation concealment on mortality to prevent the overestimation of effects of studies with inadequate or unclear allocation concealment. The studies with unclear risk were same as those identified by Stern et al. [13]; therefore, we only analyzed low risk allocation.

## 3. Results

Our search yielded 1049 citations, and 1046 records were excluded by two reviewers. Including the study by Stern et al. [13], eight studies were included. We included three additional articles that were published since 2018. After assessing the full texts of eleven articles, we identified eleven studies [12,19,20,21,22,23,24,25,26,27,28] (Figure 1). Eleven articles published between 1977 and 2022 were finally included in the study.

### 3.1. Characteristics of Studies

Four RCTs were conducted in the USA [20,23,24,26] and two RCTs were conducted in Chile [27,28]. The other studies were conducted in The Netherlands [12], Israel [25], Spain [19], and India [22]. The sample sizes ranged from 33 [24] to 281 [12] (Table 1); a total of 1128 participants were included in eleven RCTs.

### 3.2. Selection Bias

Funnel plot analyses were performed for the three main comparisons: mortality, clinical failure, and bacteremia. The funnel plots for three main comparisons were symmetrical (Figure 2). An indication that small trials are missing may be present for bacteremia in Figure 2.

### 3.3. Risk of Bias Assessment, GRADE, and Meta-Analyses

In our systematic review, no significant differences in mortality (RR 1.43, 95% CI, 0.81, 2.53, I^2^ = 0), clinical failure (RR 1.14, 95% CI, 0.86, 1.49, I^2^ = 25), and bacteremia (RR 1.32, 95% CI, 0.87, 2.01, I^2^ = 34) were observed (Figure 3). The risk of bias assessment data are graphically presented in Table 2. We also evaluated the GRADE for mortality, treatment failure, and bacteremia in the RCTs and found a low certainty of evidence (Table 3). We also analyzed the mortality, clinical failure, and bacteremia for only hematological malignancy patients, including patients who underwent stem cell transplantation as reported by de Jonge et al. [12], Ram et al. [25], Aguilar-Guisado et al. [19], and Santolaya et al. [27], and found similar results (Appendix A).

### 3.4. Sensitivity Analysis

The RR for mortality was 1.07 (95% CI 0.39–2.92) in the studies with a low risk of bias for allocation concealment (five trials), compared with an RR of 1.65 (95% CI 0.82–3.29) in the studies with an unclear risk of bias for allocation concealment (*p* = 0.51 for subgroup differences; Appendix A).

## 4. Discussion

In this systematic review, we examined the short- and long-term duration of antibiotic management for neutropenic fever in patients with hematological malignancy. We identified eleven RCTs (comprising 1128 distinct patients with FN). In our systematic review, we found that the efficacy of short-term duration of treatment may not differ statistically from that of long-term.

According to these studies, it may still be difficult for clinicians to implement a short duration of antibiotics for FN.

However, multidrug-resistant GNB, including carbapenem-resistant GNB and those caused by extended spectrum beta-lactamase-producing Enterobacteriaceae, is increasing worldwide in cancer patients [29]. Independent risk factors for CRE bloodstream infection in this study were prior β-lactam/β-lactamase inhibitor or carbapenem use. In another study, antimicrobial resistance was associated with unfavorable outcomes, such as high mortality in patients with cancer [30]. As a result, it is necessary to reduce long-term antibiotic exposure in cancer patients. 

We are still debating whether to accept the result that the long-term treatment is preferable, due to the discrepancies among the studies. The use of prophylaxis and the criteria for antibiotic discontinuation were different in each study. Regarding the different characteristics of each study, much longer durations of antibiotic treatment were reported in the study by Aguilar-Guisado et al. [19]. The type of hematological malignancies studied by de Jonge et al. [12] and Aguilar-Guisado et al. [19] were disparate. In the study by Aguilar-Guisado et al. [19], 45% of patients had acute leukemia, and approximately 27% were in induction or re-induction for prolonged neutropenia; in the study by de Jonge et al. [12], 43% of patients had multiple myeloma and about 70% of transplants were autologous, in which the neutropenic duration was shorter.

In the former study, neutrophil recovery did not resume until the patients in the short-term antibiotic group showed improved clinical symptoms; while in the latter, neutrophil recovery was early; therefore, rendering a comparison with the long-term group was difficult without a cut-off. Additionally, studies by Klaassen et al. [21] and Santolaya et al. [27] mixed the high-risk with low-risk patients. Thus, in the majority of the studies, the short-term group had very limited time to confirm culture negativity, and the long-term group may have had the desirable outcome, although the two groups were non-inferior. Although the etiology of FN remains unknown with negative culture findings, recent cell-free deoxyribonucleic acid (DNA) technology has shown that viruses and *Streptococcus viridans* are common in blood culture-negative cases [31]. We believe that if cell-free DNA is incorporated into the studies, patients with infections would be excluded from the short-term treatment group, thus, leading to more favorable outcomes.

In the context of antimicrobial regimens, de Jonge et al. [12] empirically used carbapenems, while Aguilar-Guisado et al. [19] used anti-pseudomonas beta-lactam antibodies. The regimens of other studies in the review were different from the regimen in the current guideline. However, the incidence of resistant Gram-negative-rod (GNR) strains has increased in patients with hematologic malignancies [32], but the studies included in the systematic review did not consider the ration of resistant strains. The efficacy of extended administration of beta-lactam [33] or beta-lactam + aminoglycosides [34] for FN is under investigation. These regimens are also effective against resistant GNR strains. Moreover, the small sample size of these studies also resulted in limited evidence. However, there are several ongoing RCT studies on the discontinuation of antimicrobial therapy for FN in some countries; therefore, the results are still awaited (NCT 04948463, NCT 04270786, and NCT 04637464 in the ClinicalTrials.gov registry of clinical trials).

This study had several limitations. A systematic review examines and synthesizes the information on a subject that is available in the literature; as a result, it may include some bias from the publications. We compiled trials of different designs, including RCTs and prospective non-RCTs. The study by Stern et al. [13] is supplemented by new RCTs in our study, and the authors have already communicated via email with the corresponding authors in each of the previous studies in order to learn more specifics about each investigation. We are only able to communicate with corresponding authors in each article since 2018. Although the heterogeneity of studies in our research was low, we believe that more RCTs will further improve the quality of a systematic review. Currently, several retrospective studies have been conducted on the duration of therapy for FN, which were preferable for a short course of antibiotics [35,36,37,38]. Moreover, these studies included high-risk febrile neutropenic patients in hematology. These studies support the findings of our systematic review.

## 5. Conclusions

Cancer patients should be exposed to the optimal short exposure duration of antimicrobial therapy, which benefits the implementation of antimicrobial stewardship strategies to improve the use of antimicrobials and limit multidrug resistance, as well as a short hospital stay. The evidence of each RCT is limited, and a short-term duration of beta-lactam antibiotics showed no statistically significant differences in mortality, clinical failure, and bacteremia compared with those for long-term duration antibiotics in our systematic review, possibly owing to the small number of studies, varying clinical among studies, or different study designs.

## Figures and Tables

**Figure 1 cancers-15-01611-f001:**
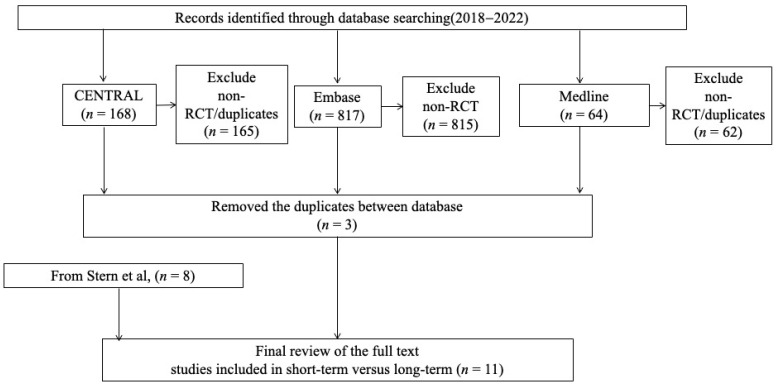
Simple identification process for eligible studies by two independent reviewers, with reference to the previous systematic review by Stern et al. [13].

**Figure 2 cancers-15-01611-f002:**
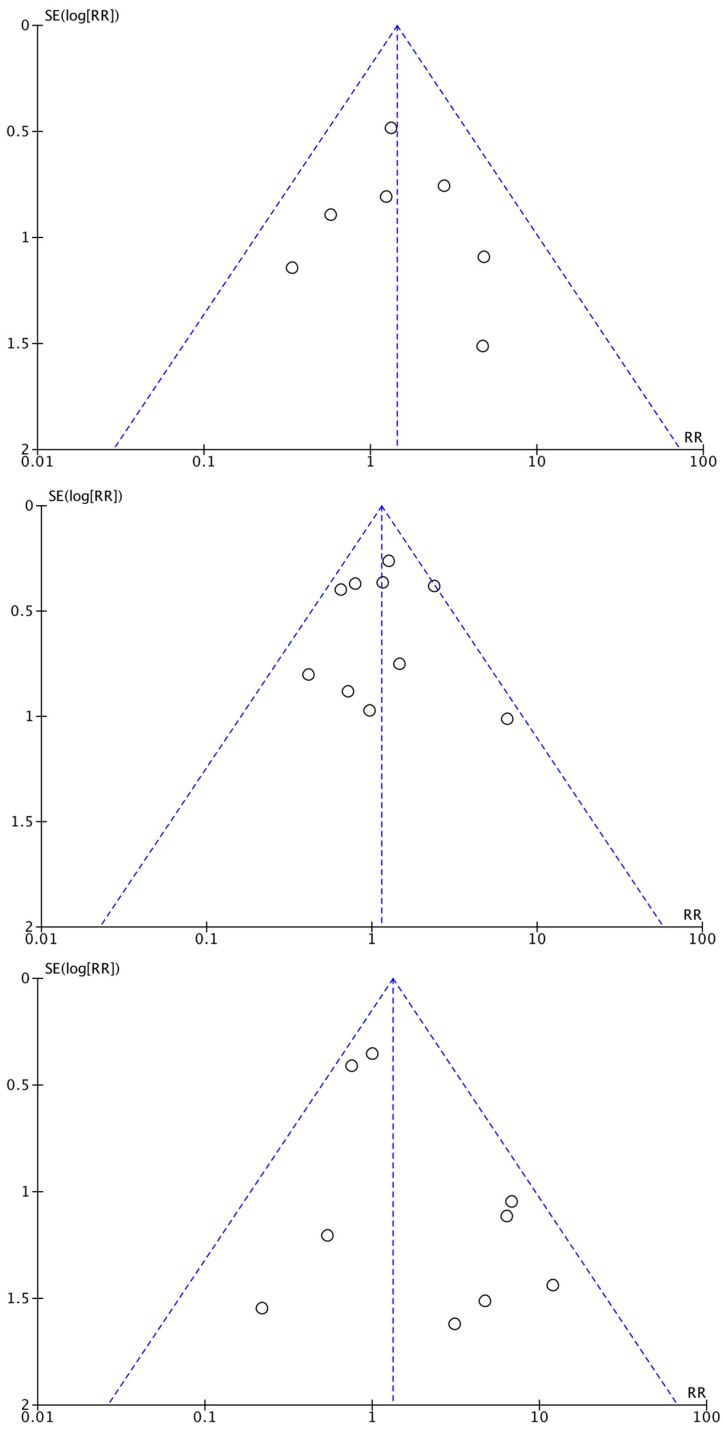
Funnel plot: mortality (**above**), clinical failure (**middle**), and bacteremia (**below**).

**Figure 3 cancers-15-01611-f003:**
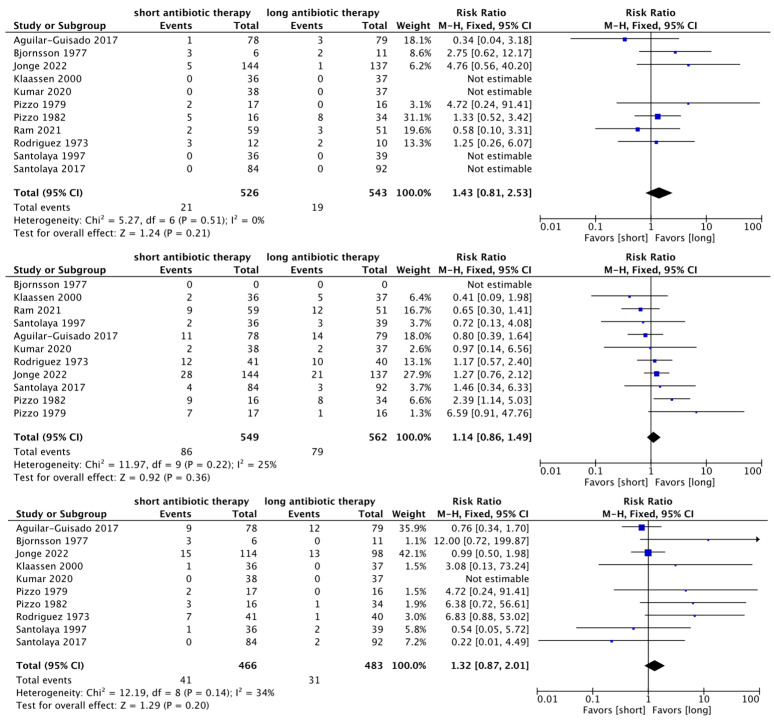
Summary of findings of short- compared with long-term duration antibiotic therapy presented as Forest plots, including the results reported by de Jonge et al. [12], Ram et al. [25], and Kumar et al. [22], and the results of the systematic review by Stern et al. [13]. Mortality (**above**), clinical failure (**middle**), and bacteremia (**below**). Total RR across 1 (left favors short, right favors long). Abbreviations: RCT, randomized control study, CI, confidence interval; RR, risk ratio [12,19,20,21,22,23,24,25,26,27,28].

**Table 1 cancers-15-01611-t001:** Clinical characteristics of the included randomized controlled trials in our systematic review.

Articles, Published Year	Published Country	Study Design/Patient Age	Study Period	Patient Characteristics, High- or Low-Risk Febrile Neutropenia/Prophylaxis in Cases of Neutropenia	Type of Beta-Lactam Antibiotics/Intervention	Number of Patients in Each Arm/Definition of Treatment Failure	Follow-Up Period
De Jonge 2022 [12]	The Netherlands	RCT, open label trial, multicenter study/median age-59 years (IQR, 52 to 65)	December 2014–July 2019	hematologic malignancy or SCT, high-risk FN/yes	IMP/CS or MEPM The antibiotics were discontinued 72 h [60–84] irrespective of the presence of fever More than 9 days until being afebrile for 5 days or neutrophil recovery	short therapy arm (*n* = 144), long-therapy arm (*n* = 137) Occurrence of either a microbiologically documented or clinically suspected carbapenem-sensitive infection; recurrence of fever from days 4–9 of empirical antibiotic treatment; or septic shock, respiratory insufficiency, or death due to any cause from day 4 until neutrophil recovery (≥0·5 × 10^9^/L)	30 days after neutrophil recovery
Ram 2021 [25]	Israel	RCT, open label trial, single center study/mean age (SD)-antibiotic stewardship strategy (intervention group, 61.2 (±12.5), standard therapy (control group), 60.6 years (±8.3)	January 2020– March 2021	HCT, CAR-T, high-risk FN/yes	PIPC/TAZ or CAZ discontinued after 48–72 h providing there was no evidence of clinical or microbiologically documented infection until recovery of counts (control group)	antibiotic stewardship strategy (*n* = 59), standard therapy (*n* = 51)/ definition of treatment success: successful response to treatment, defined as the combination of continued clinical improvement on day 5 after initiation of antibiotics, no reoccurrence of bacteremia/fever/clinical infection signs on day 5, and no need for additional therapy on days 4–5 after starting antibiotics	Not appliable
Kumar [22] 2020	India	RCT, open label trial, single center study/ mean age (SD) Arm A- 7.0 (4.0), Arm B- 8.9 (4.7)	January 2017– December 2018	all pediatric patients, aged 3–18 y with solid tumors and lymphoma leukemia/no	Cefoperazone/Sulbactam + Amikacin Prior to randomization, patient had to be afebrile for at least 24 h with a documented negative blood culture and ANC < 500, confirmation of a negative blood culture report, patients with persisting neutropenia (ANC < 500) were randomized between two arms (Arm A: stoppage of antibiotics and Arm B: short AMPC/CVA+LVFX).	Arm A (*n* = 38), Arm B (*n* = 37)/ Reoccurrence of fever	until resolution of neutropenia
Aguilar-Guisado 2017 [19]	Spain	RCT, open label trial, multicenter study/median age (IQR) short-therapy arm, 52 years (42 to 61) long-therapy arm, 54 years (39 to 63)	April 2012–May 2016	hematologic malignancy or SCT, high-risk FN/yes	antipseudomonal beta-lactam as monotherapy or combination short-until apyrexia + signs and symptoms resolution + normal vital signs long-until apyrexia + signs and symptoms resolution + normal vital signs + ANC > 500/μL	short-therapy arm (*n* = 78), long-therapy arm (*n* = 79)/ recurrent fever	28 days
Santolaya 2017 [27]	Chile	RCT, open label trial, multicenter study/mean age (SD) short therapy: 4.0 years (3 to 8) long therapy: 5.0 years (3 to 9)	July 2012–December 2015	high and low risk FN + a positive nasopharyngeal sample for a respiratory virus/yes	CTRX for low-risk FN, CAZ + AMK +/− anti-Gram-positive beta-lactam or glycopeptide for high-risk FN short therapy 3 days, stopped at randomization long therapy continuation of the same regimen until 7 days if afebrile for 24 h and CRP < 40 mg/L	short-therapy (*n* = 84), long-therapy (*n* = 92)/ development of sepsis, admission to PICU	until fever and ANC resolution
Klaassen 2000 [21]	Canada	RCT, double blind placebo-controlled trial, single center study/short therapy arm, 4.3 years long therapy arm, 4.9 years	August 1996–April 1998	low-risk FN/no	PIPC + GM 48 to 120 h after admission short term: placebo-48 to 120 h followed by placebo until 14 days or ANC > 500/μL long term: oral cloxacillin + oral cefixime	short therapy arm (*n* = 36), long therapy arm (*n* = 37)/ readmission with recurrent neutropenia	until ANC recovery
Santolaya 1997 [28]	Chile	RCT, double bind placebo- controlled trial, single center study/mean age (SD) short therapy: 6.8 years (4.3) long-therapy: 5.6 years (3.8)	January 1994– January 1996	unknown origin of FN/ regimen/no	anti-staphylococcal penicillin and a third generation cephalosporin or an AG for 3 days short therapy: no antibiotics long therapy: continuation of the same until episode of fever and neutropenia resolved	short-therapy (*n* = 36), long-therapy (*n* = 39)/documented bacterial infection + probable bacterial infection	until fever and ANC resolution
Pizzo 1982 [23]	USA	median age (range) short therapy arm: 15 years (2 to 22) long therapy arm: 16 years (2 to 25) antibacterial + amphotericin B arm: 18 years (8 to 30)	November 1975– December 1979	children with neutropenia and fever of unknown origin/no	unknown origin of FN with resolving fever after 7 days of antibiotic treatment: cefalotin + GM + carindacillin short therapy: no antibiotics long therapy: antibacterial arm: continuation of the same regimen until afebrile for ≥24 h and ANC > 500, antibacterial + amphotericin B arm: continuation of the same regimen + amphotericin B (0.5 mg/kg/d, IV)	short therapy arm (*n* = 16) long therapy arm (*n* = 16) antibacterial + amphotericin B arm (*n* = 18)/any infectious complication	until fever and ANC resolution
Pizzo 1979 [24]	USA	RCT, open label, single center study/median age (range) short therapy arm: 14 years (2 to 33) long therapy arm: 15 years (1 to 30)	November 1975– February 1978	unknown origin of FN with resolving fever after 7 days of antibiotic treatment/no	cefalotin + GM + carindacillin short-therapy arm: no antibiotics Long-therapy arm: continuation of the same regimen Day start (day of randomization): day 7 from admission until afebrile for ≥24 h and ANC > 500	short therapy arm (*n* = 17) long therapy arm (*n* = 16)/ recurrence of fever	30 days after fever and ANC resolution
Bjornsson 1977 [20]	USA	RCT, open label, single center study/mean age (SD) short therapy arm: 42.5 years (±11.8) long therapy arm: 43.45 years (±16.5)	June 1975– May 1976	unknown origin of unresolving FN/no	after 3 days of antibiotic treatment/ carbenicillin + cephalothin + gentamicin short: no antibiotics long: carbenicillin + cephalothin + gentamicin + CLDM or CL	short therapy arm (*n* = 6) long therapy arm (*n* = 11)/	4 weeks
Rodriguez 1973 [26]	USA	RCT, open label, single center study/median 33 (range, 15–80) years	July 1970– December 1971	unknown origin of FN/no	cefalotin + carindacillin short therapy: 4 days long therapy: 10 days of additional therapy (total 14 days) or 5 days after becoming afebrile, whichever was longer	short therapy non-resolving fever (*n* = 11), resolving fever (*n* = 30), long therapy: non resolving fever (*n* = 14) resolving fever (*n* = 26)/ infection is cause of fever	Not applicable

Abbreviations: CTRX, ceftriaxone; CAZ, ceftazidime; MEPM, meropenem; PIPC, piperacillin; PIPC/TAZ, piperacillin/tazobactam; CLDM, clindamycin; CL, Chloramphenicol; GM, gentamycin; AG, aminoglycoside; AMK, amikacin; FN, febrile neutropenia; RCT, randomized control trials; N/A, not available; ANC, absolute neutrophil count; IQR, interquale range; SD, standard deviation.

**Table 2 cancers-15-01611-t002:** Summary of risk of bias in all the randomized controlled trials including the results reported by de Jonge et al. [12], Ram et al. [25], and Kumar et al. [22], and the results of the systematic review by Stern et al. [13]. The risk of bias included randomization sequence, concealment, blinding of participant and clinician, incomplete outcome data, selective reporting, and others. The color of risk of bias: green, low risk of bias; yellow, unclear risk of bias; red, high risk of bias.

		Randomization Sequence	Concealment	Blinding of Participant and Clinician	Blinding of Outcome Assessment	Incomplete Outcome Data	Selective Reporting	Others
our analysis	de Jonge 2022 [12]	low	low	high	low	low	low	low
	Kumar 2020 [22]	low	low	high	low	low	low	low
	Ram 2021 [25]	low	low	high	unclear	low	unclear	unclear
Stern analysis	Aguilar-Guisado 2017 [19]	low	low	high	high	low	low	low
	Bjornsson 1977 [20]	unclear	unclear	high	high	low	high	low
	Klaassen 2000 [21]	low	low	low	low	low	low	low
	Pizzo 1979 [24]	unclear	unclear	high	high	low	high	unclear
	Pizzo 1982 [23]	unclear	unclear	high	high	low	high	unclear
	Rodriguez 1973 [26]	unclear	unclear	high	high	low	high	unclear
	Santolaya 1997 [28]	unclear	unclear	high	high	low	high	unclear
	Santolaya 2017 [27]	unclear	unclear	high	low	low	high	low
	low (%)	45.5	45.5	9.1	36.4	100.0	36.4	54.5
	unclear (%)	54.5	54.5	0.0	9.1	0.0	9.1	45.5
	high (%)	0.0	0.0	90.9	54.5	0.0	54.5	0.0

**Table 3 cancers-15-01611-t003:** Summary of findings for mortality, clinical failure, and bacteremia in our systematic review. Short duration of antibiotic treatment compared with long duration of antibiotics treatment.

Summary of Findings						
Short compared with long antibiotic therapy duration for febrile neutropenia						
Patient or population: febrile neutropenia						
Setting:						
Intervention: short						
Comparison: long						
Outcomes	Anticipated absolute effects * (95% CI)		Relative effect (95% CI)	№ of participants (studies)	Certainty of the evidence (GRADE)	Comments
	Risk with long	Risk with short				
mortality	35 per 1000	50 per 1000 (28 to 89)	RR 1.43 (0.81 to 2.53)	1069 (11 RCTs)	⊕⊕◯◯ Low ^a,b^	
treatment failure	141 per 1000	160 per 1000 (121 to 209)	RR 1.14 (0.86 to 1.49)	1111 (10 RCTs)	⊕⊕◯◯ Low ^a,b,c^	
bacteremia	64 per 1000	85 per 1000 (56 to 129)	RR 1.32 (0.87 to 2.01)	949 (10 RCTs)	⊕⊕◯◯ Low ^a,b^	
* The risk in the intervention group (and its 95% confidence interval) is based on the assumed risk in the comparison group and the relative effect of the intervention (and its 95% CI).						
CI: confidence interval; RR: risk ratio						
GRADE Working Group grades of evidence						
High certainty: we are very confident that the true effect lies close to that of the estimate of the effect.						
Moderate certainty: we are moderately confident in the effect estimate: the true effect is likely to be close to the estimate of the effect, but there is a possibility that it is substantially different.						
Low certainty: our confidence in the effect estimate is limited: the true effect may be substantially different from the estimate of the effect.						
Very low certainty: we have very little confidence in the effect estimate: the true effect is likely to be substantially different from the estimate of the effect.						

Explanations: ^a^. methods of randomization and allocation concealment were unclear in most studies. Most studies were unblinded; ^b^. effect estimate overlapping no effect with wide confidence interval. ^c^. variable and inconsistent definition of clinical failure across studies.

## Data Availability

The data presented in this study are available on request from the corresponding author (Kazuhiro Ishikawa).

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
