# Peer review of "Systematic Review of the Short-Term versus Long-Term Duration of Antibiotic Management for Neutropenic Fever in Patients with Cancer"

_cancers, 2023, doi:10.3390/cancers15051611_

Round 1
Reviewer 1 Report
Ishikawa et al. performed systematic review and meta analysis of 11 RCTs comparing short vs. long antibiotic duration in patients with hematologic diseases.
The study was well designed and overall followed the Cochrane recommendations.
I have 2 major comments -
1. I think that conclusion from this analysis is that short duration antibiotic is not inferior to long duration. As long duration antibiotic is the standard practice in majority of institutions and CI cross 1 in all comparisons, my conclusion is that these 2 options are comparable. There are pro and cons unique to each option and this has been discussed by the authors.
2. The authors did not perform sensitivity analysis to test whether risk of bias impact conclusions. In addition, no subgroup analysis was performed to test whether certain option is better than the other in specific subgroups of patients (allo/AML induction vs. other patients with "lower" risk FN. Also analyzing secondary outcome like organ toxicity, CDT infection will be of interest.
In addition, please check the first author name in the Funnel Plots and in Table 1 (the Header is missing).
Author Response
I have 2 major comments -
1. I think that conclusion from this analysis is that short duration antibiotic is not inferior to long duration. As long duration antibiotic is the standard practice in majority of institutions and CI cross 1 in all comparisons, my conclusion is that these 2 options are comparable. There are pro and cons unique to each option and this has been discussed by the authors.
Response>Thank you for your feedback. I agree with you. The risk ratio of the total studied is slightly favorable for extended duration of treatment, the CI across. Because shorter therapy was safer, we consider the shorter duration of therapy for the treatment strategy of FN. Specifically clinically stable, with no fever and no signs of symptoms are safer in shorter therapy of FN.
2. The authors did not perform sensitivity analysis to test whether risk of bias impact conclusions. In addition, no subgroup analysis was performed to test whether certain option is better than the other in specific subgroups of patients (allo/AML induction vs. other patients with "lower" risk FN. Also analyzing secondary outcome like organ toxicity, CDT infection will be of interest.
Response>Thank you for your feedback. I analyzed the mortality, clinical failure, and bacteremia in high-risk FN including hematological cancer and stem cell transplantation patients described Line 238-241. These results were similar to the total group. Regarding the adverse events, these data are limited. Therefore, Stern et al did not analyze them. In diarrhea, N. A. de Jonge showed only 2% in shorter therapy. In Clostridium difficile colitis, shorter therapy has 2/78 in the experimental course of Agui-lar-Guisado. In renal failure, the longer therapy of treatment has 4/177(2%)(N. A. de Jonge, Agui-lar-Guisado). The result was a renal failure (RR 0.18, 95% CI, 0.02, 1.58, I2 = 0).
We also performed a sensitivity analysis to assess the effect of allocation concealment on the bacteremia to prevent the overestimation of effects of studies with inadequate or unclear allocation concealment in Line 185-189, 233-237.
In addition, please check the first author name in the Funnel Plots and in Table 1 (the Header is missing).
Response>Thank you for your feedback. I checked it.
Reviewer 2 Report
· A brief summary The authors performed a systematic review comparing short versus extended antibiotic treatment of haematology patients with fever during neutropenia and used clinical failure, occurrence of bacteraemia and death due to any cause and the main outcomes of interest. The authors used the previously performed Cochrane Review by Stern et al. (PMID 30605229) as framework for their review and added three more studies in this work. This review included studies with both paediatric and adult haematology (and oncological?) patients with neutropenia (defined as a neutrophil count <500 cells/uL) and fever (temperature >= 38.3 degrees Celsius or sustained >=1 hour >= 38.0 degrees Celsius), which both are in line with commonly used international definitions. The authors excluded neonatal patients and patients with microbiologically documented infections. This systematic review included 11 studies between 1997 and 2022 including 1128 participants. No significant differences were observed for the outcomes mortality, clinical failure and bacteraemia. The overall quality of evidence was considered as very low. The authors conclude that no strong conclusions can be drawn due to the very low quality of evidence.
Review
The authors performed an update of an existing Cochrane Review by including three more studies. The conclusion that no differences were found is therefore not really surprising. This also limits the scientific novelty of this work. It would however be useful to limit the population to the haematological population, because no previous review has been performed in this high-risk population. The title already suggested this, but in the study it seemed that the authors decided to include a broad oncological population, which is considered be at less risk of complications. The introduction starts with a good introduction to the reader but lacks a clear structure and needs thorough revision. The methods used for this review are in accordance with international standards, seem to have been performed well. In the Discussion, the authors should better describe how their results fit in the broader body of literature (i.e. large retrospective studies and guidelines), and perhaps how their results should be incorporated in clinical practice or stewardship programs. The heterogeneity of study methodology hampers the authors from drawing any strong conclusions, so they should maybe also elaborate more about this in the Discussion.
Specific comments
· Simple summary/Abstract
o Please add the following:
§ The objective is not explicitly reported, but given the selected endpoints, the authors probably intended to investigate the safety of early antibiotic discontinuation in febrile neutropenia.
§ Lines 26-28: Please add ‘randomised’ to clinical trials.
§ Lines 26: The date of the database search is not described the abstract.
§ Please include the number of included studies and participants in the abstract.
§ Consider to include the direction of effect to clarify that the risk ratios favour extended treatment.
o Lines 32-33: “We could make no strong conclusions on the safety and efficacy of antimicrobial discontinuation before neutropenia resolution with FN”. This sentence is nearly the same as the conclusion of the manuscript by Stern et al. I encourage the authors to rephrase this sentence.
· Introduction:
o The authors describe the well-known advantages of restrictive antibiotic administration, but the reasons for extended antibiotic treatment in many centres are not addressed. This is mainly to the clinical dilemma that infection can have severe consequences to these immunocompromised patients. Also, blood cultures are considered less reliable as these are drawn during (prophylactic) antibiotic treatment. Lastly, many patients have persistent fever even though no infectious cause is identified. This is generally considered to be due to chemotherapy-induced mucositis. The authors don’t address that in literature and therefore also in the included articles in their manuscript, generally two different de-escalation approaches are being used, namely fixed short duration regardless of fever presence or fever-guided discontinuation. These different approaches have consequences for the length of antibiotic treatment and therefore for the interpretation of the studies. The introduction would benefit from a full outline of the subject. Also, the introduction lacks a clear structure and needs thorough revision.
o Lines 52-53, In two large clinical trials[...]: the authors state that current treatment duration in haematology patients is longer than in many other serious infections in immunocompetent patients. The optimal treatment duration of infections in immunocompromised patients is still matter of debate, but generally treatment length is extended due to impaired cellular and humoral immunity. It is not clear what point the authors would like to make in the broader light of this paragraph.
o Line 68: The authors state the study of De Jonge et al. does not show any differences in safety and mortality outcomes, but this study showed increased 30-day post-neutrophil recovery mortality as well as increased serious adverse events.
· Methods
o Line 85 “Effectiveness”: the mentioned outcomes clinical failure, bacteraemia and mortality rather refer to safety than effectiveness. “Safety” would therefore be a more suitable term.
o Lines 96 (PROSPERO): A few discrepancies with the published protocol on PROSPERO were identified:
§ Fever was defined as on tympanic measurement of 38.3 degrees Celsius or above or persistently above 38.0 degrees Celsius. Initially the fever was defined as 37.5. What was the reason for this change and how did this affect the selection of selected articles?
§ The authors describe that they excluded both clinically and microbiologically documented infections (MDI). In the manuscript, only MDIs were excluded. (Lines 102)
o Line 125 “type of cancer (, solid tumour…): the title of the manuscript suggests that the study population only involves haematological malignancies, but throughout the manuscript the authors suggest that also patients with solid tumours are included. The authors should clearly define which the population and if patients with solid tumours are also included, the title should reflect this.
o Paragraph 2.5 “Data extraction”: many of the included studies were published some decades ago. The others suggest that there was missing data (lines 144-145). How did the authors handle missing data that could not be retrieved from the original authors?
o What was the reason that antibiotic treatment length was not added as secondary endpoint? I would be interesting to show the differences of treatment duration between different de-escalation strategies.
· Results
o Figure 1: Please add a breakdown of reasons for excluding studies in the flowchart.
o Lines 199-200: The geographic distribution of 6 of 11 studies is described. Please complete this by adding the other 5 studies.
o Table 1:
§ Please add if antibiotic prophylaxis was used in the studies as this is an important confounder.
§ Given the variety of definitions for “clinical failure” I would encourage the authors to add this to Table 1.
§ The description of the average age (7th column) seems a bit redundant as this is already described in column 3.
§ “Average”, do the authors intend to use “mean”?
§ Santolaya et al 2017 (ref 16): the authors excluded studies with microbiologically documented infections, but this study included patients with positive nasopharyngeal swabs for a respiratory virus. The authors should explain this discrepancy or exclude this study.
o Figure 3: the denominator of the study of De Jonge et al is pane C (mortality) is 114 and 98 whereas this is 144 and 137 in the panes above. Please verify if these numbers are correct.
o Page 19: the authors grade the quality of evidence for mortality as very low for mortality, while this was graded as low in the study by Stern et al. Three additional studies were added to the existing evidence, of all of which were RCTs. Also, from the summary of the risk of bias (Figure 4) it can be concluded that many of the studies had less risk of bias than the analysis by Stern et al. Could the authors explain why the certainty of evidence for the endpoint mortality was lower than in the study of Stern et al. (Very low vs low)?
· Discussion
o Lines 262-263: In our systematic…: The authors state that long-term duration of treatment is preferable for all outcomes. What finding justifies this conclusion? None of the shows a statistically significant difference. Conversely, an argument could be made that, in the absence of a statistically difference, short treatment is preferred due to its other advantages.
o Lines 273-279: The studies of De Jonge et al. and Aguilar-Guisado et al. were not only disparate in population. Also the use of antibacterial prophylaxis and the criteria for antibiotic discontinuation were different, thereby leading to much longer duration of antibiotic treatment lengths in the study of Aguilar-Guisado. The authors might also bring up that the bacteraemia episodes in the study of De Jonge et al were mostly not carbapenem-susceptible and could therefore not have been prevented by extending the empirical treatment. This result was not reproduced in the study of Aguilar-Guisado et al.
o Lines 284-286: The authors suggest that the limited time to blood culture positivity before antibiotic discontinuation may have influenced the results. I doubt this argument as the median time to positivity in febrile neutropenia is 12 hours, and 92% of positive blood cultures is returned to the clinician in the first 24 hours. (PMID 30096417) However, I agree that thorough diagnostic investigations should be performed to ensure that no infection is underdiagnosed.
o I encourage the authors to also discuss the results of this systematic review in the broader scope of studies that were performed, but were not included in this review. Although these studies were retrospective, some of them have quite a large sample size (Schauwvlieghe et al; Le Clech et al, Le Martire et al, Niessen et al.).
o The authors performed an update of the previously performed systematic review of Stern et al. The authors should describe how this update influences the interpretation of their results in comparison to the study by Stern et al.
o Line 314 “Show any difference” please add ‘statistically significant”.
o Line 317 “Furthermore, cancer patients are expected to have optimal short exposure to antimicrobial therapy”. What do the authors mean by this? The clinical dilemma remains that empirical antibiotics should be stopped in absence of infections to prevent antibiotic resistance and selection, but on the other hand patients should not be undertreated. Given this vulnerable population, this balance has so far always been in favour of extended treatment to be on the safe side.
o Lines 320 “Evidential”: what do the authors mean by this? All the RCTs contributed to the evidence, but no difference has hitherto been observed. Or do the authors mean that quality of the methodology should be increased such as placebo-controlled trials?
· Stylistic comments
o Lines 187: “more three” should be “three more”
o References to the authors of the included studies are inconsistently referred to by only their family name or first name and family name. Please adjust this for consistency.
o Table 1: “Netherland”, should be “the Netherlands”.
o Figure 2 subscript: bellow, should be below.
o Figure 3: please include titles so the reader can easily see what the outcome of the analysis is. Favour is the preferred spelling outside the U.S.A., but this manuscript is written in American spelling, so “favor” would be more appropriate.
o Line 213: bactereua, should be bacteremia
o Line 214: “among the bacteremia”(?)
o Line 283: “Klassen” should be “Klaassen”.
Author Response
A brief summary The authors performed a systematic review comparing short versus extended antibiotic treatment of haematology patients with fever during neutropenia and used clinical failure, occurrence of bacteraemia and death due to any cause and the main outcomes of interest. The authors used the previously performed Cochrane Review by Stern et al. (PMID 30605229) as framework for their review and added three more studies in this work. This review included studies with both paediatric and adult haematology (and oncological?) patients with neutropenia (defined as a neutrophil count <500 cells/uL) and fever (temperature >= 38.3 degrees Celsius or sustained >=1 hour >= 38.0 degrees Celsius), which both are in line with commonly used international definitions. The authors excluded neonatal patients and patients with microbiologically documented infections. This systematic review included 11 studies between 1997 and 2022 including 1128 participants. No significant differences were observed for the outcomes mortality, clinical failure and bacteraemia. The overall quality of evidence was considered as very low. The authors conclude that no strong conclusions can be drawn due to the very low quality of evidence.
Review
The authors performed an update of an existing Cochrane Review by including three more studies. The conclusion that no differences were found is therefore not really surprising. This also limits the scientific novelty of this work. It would however be useful to limit the population to the haematological population, because no previous review has been performed in this high-risk population. The title already suggested this, but in the study it seemed that the authors decided to include a broad oncological population, which is considered be at less risk of complications. The introduction starts with a good introduction to the reader but lacks a clear structure and needs thorough revision. The methods used for this review are in accordance with international standards, seem to have been performed well. In the Discussion, the authors should better describe how their results fit in the broader body of literature (i.e. large retrospective studies and guidelines), and perhaps how their results should be incorporated in clinical practice or stewardship programs. The heterogeneity of study methodology hampers the authors from drawing any strong conclusions, so they should maybe also elaborate more about this in the Discussion.
Specific comments
· Simple summary/Abstract
o Please add the following:
§ The objective is not explicitly reported, but given the selected endpoints, the authors probably intended to investigate the safety of early antibiotic discontinuation in febrile neutropenia.
Response>Thank you for your feedback. I added, “We intended to investigate the safety of early antibiotic discontinuation in FN.” in Line 24-25.
§ Lines 26-28: Please add ‘randomised’ to clinical trials.
Response>Thank you for your feedback. I added the “randomized” in Line 27.
§ Lines 26: The date of the database search is not described the abstract.
Response>Thank you for your feedback. I added the data of the database search “on September 30, 2022” in Line 26.
§ Please include the number of included studies and participants in the abstract.
Response>Thank you for your feedback. I added the “We identified eleven RCTs (comprising 1,128 distinct patients with FN) from 1977-2022.” in Line 29-30
§ Consider to include the direction of effect to clarify that the risk ratios favour extended treatment.
Response> Thank you for your feedback. I added the “indicating that the efficacy short-term treatment may not differ statistically different from that of long-term treatment.” in Line 32-33.
Lines 32-33: “We could make no strong conclusions on the safety and efficacy of antimicrobial discontinuation before neutropenia resolution with FN”. This sentence is nearly the same as the conclusion of the manuscript by Stern et al. I encourage the authors to rephrase this sentence.
Response> Thank you for your feedback. As I discussed later, a Risk ratio of mortality, clinical failure, and bacteremia cross 1. There is no significant difference between short therapy and long therapy. I change the sentence, “Regarding patients with FN, our findings provide weak conclusions regarding the safety and efficacy of antimicrobial discontinuation prior to neutropenia resolution. ” in Line 33-35.
· Introduction:
o The authors describe the well-known advantages of restrictive antibiotic administration, but the reasons for extended antibiotic treatment in many centres are not addressed. This is mainly to the clinical dilemma that infection can have severe consequences to these immunocompromised patients. Also, blood cultures are considered less reliable as these are drawn during (prophylactic) antibiotic treatment. Lastly, many patients have persistent fever even though no infectious cause is identified. This is generally considered to be due to chemotherapy-induced mucositis. The authors don’t address that in literature and therefore also in the included articles in their manuscript, generally two different de-escalation approaches are being used, namely fixed short duration regardless of fever presence or fever-guided discontinuation. These different approaches have consequences for the length of antibiotic treatment and therefore for the interpretation of the studies. The introduction would benefit from a full outline of the subject. Also, the introduction lacks a clear structure and needs thorough revision.
Lines 52-53, In two large clinical trials[...]: the authors state that current treatment duration in haematology patients is longer than in many other serious infections in immunocompetent patients. The optimal treatment duration of infections in immunocompromised patients is still matter of debate, but generally treatment length is extended due to impaired cellular and humoral immunity. It is not clear what point the authors would like to make in the broader light of this paragraph.
Response> Thank you for your feedback. I change the sentence in order to clarify the shorter duration of antimicrobial therapy is important for cancer patients in line 60-69.
“In general, the shorter duration of antimicrobials reduces the development of multidrug-resistant organisms, antimicrobial-induced adverse drug events such as antibiotic-associated diarrhea, Clostridioides difficile infection, allergy reaction and nephrotoxicity, and hospital length of stay. In hematological cancer patients, the reduction of antimicrobial therapy durations leads to less unnecessary antimicrobial exposure, which is an additional benefit for patients with hematological cancer that has not previously been described in this vulnerable population.
Line 68: The authors state the study of De Jonge et al. does not show any differences in safety and mortality outcomes, but this study showed increased 30-day post-neutrophil recovery mortality as well as increased serious adverse events.
Response> Thank you for your feedback. I discussed the study of De Jonge et al is the worse outcome in mortality and adverse events but still within the non-inferiority margin of 10%. This study included many multiple myelomas in which the duration of neutropenia is shorter, and the cut-off for the shorter duration is 72 hours, nevertheless the clinical status. This connected to the worse outcome, I think.
· Methods
Line 85 “Effectiveness”: the mentioned outcomes clinical failure, bacteraemia and mortality rather refer to safety than effectiveness. “Safety” would therefore be a more suitable term.
Response> Thank you for your feedback. I changed the safety from the effectiveness in Line 92.
Lines 96 (PROSPERO): A few discrepancies with the published protocol on PROSPERO were identified:
Response> Thank you for your feedback. I have some modifications to my manuscript from the data of registration in PROSPERO. I am now updating the protocol of Prospero.
§ Fever was defined as on tympanic measurement of 38.3 degrees Celsius or above or persistently above 38.0 degrees Celsius. Initially the fever was defined as 37.5. What was the reason for this change and how did this affect the selection of selected articles?
Response> Thank you for your feedback. I defined the body temperature of FN from IDSA, ECIL Guideline. I will modify the protocol of PROSPERO. It was referred from Stern et al.
§ The authors describe that they excluded both clinically and microbiologically documented infections (MDI). In the manuscript, only MDIs were excluded. (Lines 102)
Response> Thank you for your feedback. I changed the exclusion criteria for this study.
Clinically and microbiologically documented infections were excluded in Line 106-107.
Line 125 “type of cancer (, solid tumour…): the title of the manuscript suggests that the study population only involves haematological malignancies, but throughout the manuscript the authors suggest that also patients with solid tumours are included. The authors should clearly define which the population and if patients with solid tumours are also included, the title should reflect this.
Response>Thank you for your feedback. As you mentioned, some study includes solid tumor patients. So, I replaced the hematological patient with cancer patient in title and Line 27-28.
Paragraph 2.5 “Data extraction”: many of the included studies were published some decades ago. The others suggest that there was missing data (lines 144-145). How did the authors handle missing data that could not be retrieved from the original authors?
Response>Thank you for your feedback. As you know, a systematic review has some limitations because several articles have missing data. So, the author in the systematic review communicated with the corresponding author in the previous RCT. But we sometime could not communicate with the author because the study was too old. Therefore, I referred to the value of the missing data from Stern et al. I could only contact the authors of 3 new articles.
What was the reason that antibiotic treatment length was not added as secondary endpoint? I would be interesting to show the differences of treatment duration between different de-escalation strategies.
Response>Thank you for your feedback. At first, we would like to estimate the difference in the duration of empirical antimicrobials, but this duration was determined in each study. So we could not analyze the data. The following is the reference of the data for the duration of antimicrobials.
long (day) short(day)
Nick A de Jonge 2022 8 day 3 day
Santolaya 2017 7 day 3 day
· Results
Figure 1: Please add a breakdown of reasons for excluding studies in the flowchart.
Response>Thank you for your feedback. I added a breakdown of reasons for excluding studies. I modified the sentence in the manuscript in Line 191 and the flowchart (figure 1).
Lines 199-200: The geographic distribution of 6 of 11 studies is described. Please complete this by adding the other 5 studies.
Response> Thank you for your feedback. I added the sentence “The other studies were conducted in the Netherlands[4], Israel[14], Spain[8], and India[11].” in Line 202-203.
o Table 1:
§ Please add if antibiotic prophylaxis was used in the studies as this is an important confounder.
Response>Thank you for your feedback. I added the new column if the included patients took prophylaxis in table 1.
§ Given the variety of definitions for “clinical failure” I would encourage the authors to add this to Table 1.
Response> Thank you for your feedback. I added the definition of clinical failure in table 1.
§ The description of the average age (7th column) seems a bit redundant as this is already described in column 3.
Response> Thank you for your feedback. I summarized the explanation of the age in column 3.
§ “Average”, do the authors intend to use “mean”?
Response> Thank you for your feedback. I modified mean age in the age of column.
§ Santolaya et al 2017 (ref 16): the authors excluded studies with microbiologically documented infections, but this study included patients with positive nasopharyngeal swabs for a respiratory virus. The authors should explain this discrepancy or exclude this study.
Response> Thank you for your feedback. Santolaya et al. 2017 included respiratory virus infection. We think that respiratory virus infection does not need antimicrobials. So, respiratory virus may not for clinician affect the strategy of duration of antimicrobials. So, we included this study.
Figure 3: the denominator of the study of De Jonge et al is pane C (mortality) is 114 and 98 whereas this is 144 and 137 in the panes above. Please verify if these numbers are correct.
Response> Thank you for your feedback. I calculated the risk ratio of plane C (bacteremia) using per-protocol analysis (114 and 98).
Page 19: the authors grade the quality of evidence for mortality as very low for mortality, while this was graded as low in the study by Stern et al. Three additional studies were added to the existing evidence, of all of which were RCTs. Also, from the summary of the risk of bias (Figure 4) it can be concluded that many of the studies had less risk of bias than the analysis by Stern et al. Could the authors explain why the certainty of evidence for the endpoint mortality was lower than in the study of Stern et al. (Very low vs low)?
Response> Thank you for your feedback. Our team discussed the bias risk of publication bias. We still have many unclear risks of bias in the concealment. So, we define serious bias risk. But we modified other risks. We concluded that there is low certainty in the mortality in Table 2, Line 30, 227.
· Discussion
Lines 262-263: In our systematic…: The authors state that long-term duration of treatment is preferable for all outcomes. What finding justifies this conclusion? None of the shows a statistically significant difference. Conversely, an argument could be made that, in the absence of a statistically difference, short treatment is preferred due to its other advantages.
Response> Thank you for your feedback. Our results are not statically different in both arms.
We modified our conclusion in Line 32-33, 276-278.
Lines 273-279: The studies of De Jonge et al. and Aguilar-Guisado et al. were not only disparate in population. Also the use of antibacterial prophylaxis and the criteria for antibiotic discontinuation were different, thereby leading to much longer duration of antibiotic treatment lengths in the study of Aguilar-Guisado. The authors might also bring up that the bacteraemia episodes in the study of De Jonge et al were mostly not carbapenem-susceptible and could therefore not have been prevented by extending the empirical treatment. This result was not reproduced in the study of Aguilar-Guisado et al.
Response> Thank you for your feedback. I agree with you. I added the prophylaxis in the table of studies and added your comment in Line 289-292.” The use of prophylaxis and the criteria for antibiotic discontinuation were different in each study. “For the different characteristics of each study, much longer duration of antibiotic treatment lengths in the study of Aguilar-Guisado et al.[8]”
Lines 284-286: The authors suggest that the limited time to blood culture positivity before antibiotic discontinuation may have influenced the results. I doubt this argument as the median time to positivity in febrile neutropenia is 12 hours, and 92% of positive blood cultures is returned to the clinician in the first 24 hours. (PMID 30096417) However, I agree that thorough diagnostic investigations should be performed to ensure that no infection is underdiagnosed.
Response> Thank you for your feedback. I agree that searching for the source of febrile neutropenia is very difficult because the bacteremia is only 15-35%, but the unknown origin is 45-50%[ PMID: 23975584]. As I discussed, cell-free deoxyribonucleic acid revealed the culture-negative but true infection in FN. Also respiratory virus infection is common for FN, multiplex PCR(bioMérieux, Filmarray) is helpful. So, using this technology, We could search for the cause of FN, and the percentage of unknown of origins will be reduced.
I encourage the authors also to discuss the results of this systematic review in the broader scope of studies that were performed but were not included in this review. Although these studies were retrospective, some of them have quite a large sample size (Schauwvlieghe et al.; Le Clech et al., Le Martire et al., Niessen et al.).
Thank you for your comment. The studies of Schauwvlieghe(PMID: 33997746), Le Clech(PMID: 29451055, ), Le Martire(PMID: 30051357), Niessen (PMID: 32460887), as you pointed out, were retrospective study, but these results are preferable for a short course of antibiotics. Also, these studies included more high-risk febrile neutropenic patients in hematology than our systematic review. I added your feedback in Line 330-333.
The authors performed an update of the previously performed systematic review of Stern et al. The authors should describe how this update influences the interpretation of their results in comparison to the study by Stern et al.
Response>Thank you for your comment. In our systematic review, we added the two studies of high-risk hematological malignancy patients(Ram, Jonge). Stern et al. included only Aguilar-Guisado in high-risk FN, and other studies' profile is unknown. But short duration was not significantly different from the long course in mortality, treatment failure, and bacteremia between our results and Stern et al.
Line 314 “Show any difference” please add ‘statistically significant”.
Response>Thank you for your feedback. I added the statistically significant in Line 339.
Line 317 “Furthermore, cancer patients are expected to have optimal short exposure to antimicrobial therapy”. What do the authors mean by this? The clinical dilemma remains that empirical antibiotics should be stopped in absence of infections to prevent antibiotic resistance and selection, but on the other hand patients should not be undertreated. Given this vulnerable population, this balance has so far always been in favour of extended treatment to be on the safe side.
Response>Thank you for your feedback; this study showed that short-course therapy does not inferior to long therapy, as you pointed out, I modified the conclusion.
Lines 320 “Evidential”: what do the authors mean by this? All the RCTs contributed to the evidence, but no difference has hitherto been observed. Or do the authors mean that quality of the methodology should be increased such as placebo-controlled trials?
Response>Thank you for your feedback. I modified our conclusion that short therapy may be safer than long therapy.
· Stylistic comments
Lines 187: “more three” should be “three more”
Response>Thank you for your feedback. I changed three additonal from more three in Line 192.
References to the authors of the included studies are inconsistently referred to by only their family name or first name and family name. Please adjust this for consistency.
Response>Thank you for your feedback. I changed the authors of included studies in the studies of table and figure.
Table 1: “Netherland”, should be “the Netherlands”.
Response>Thank you for your feedback. I changed Netherlands in Table 1.
Figure 2 subscript: bellow, should be below.
Response>Thank you for your feedback. I changed below in figure 2.
Figure 3: please include titles so the reader can easily see what the outcome of the analysis is. Favour is the preferred spelling outside the U.S.A., but this manuscript is written in American spelling, so “favor” would be more appropriate.
Response>Thank you for your feedback. I added the title in figure 3. I changed favor from favour in all the figures.
Line 213: bactereua, should be bacteremia
Response>Thank you for your feedback. I modified the “bacteremia” from “bactereua”.
Line 214: “among the bacteremia”(?)
Response>Thank you for your feedback. I modified the “in the bacteremia of Figure 2” from “among the bacteremia”.
o Line 283: “Klassen” should be “Klaassen”.
Response>Thank you for your feedback. I modified the “Klaassen” from Klassen”.
Round 2
Reviewer 1 Report
My comments have been nicely addressed and I recommend publication.